# Enhancement of Filtration Performance Characteristic of Glass Fiber-Based Filter Media, Part 2: Chemical Modification with Surface-Active Treatment

**DOI:** 10.3390/ma17112720

**Published:** 2024-06-03

**Authors:** Laura Weiter, Stephan Leyer, John K. Duchowski

**Affiliations:** 1Faculty of Science, Technology and Communication, University of Luxembourg, 4365 Luxembourg, Luxembourg; laura.weiter@hydac.com (L.W.); stephan.leyer@uni.lu (S.L.); 2HYDAC FluidCareCenter® GmbH, 66280 Sulzbach, Germany

**Keywords:** chemical modification, surface treatment, work of adhesion

## Abstract

Standard glass fiber filter media were chemically modified with suitably chosen surface-active agents. The aim of these modifications was to improve the three fundamental filtration performance characteristics, namely, to increase the separation efficiency, reduce the differential pressure (∆P) and increase the dirt holding capacity (DHC). The increase in separation efficiency was considered quantitatively in terms of changes in the work of adhesion between the contaminant and the modified media substrate derived from the contact angle measurements. The experimental confirmation of this behavior was demonstrated by an improved separation efficiency especially for particles in the smaller size ranges, well below the mean porosity of the original substrate. In addition, the effect of different surface modifications, especially those of the opposite ends of the surface energy values, has clearly manifested itself in the experimental results of separation efficiency derived from the multipass evaluations. Collectively, the obtained contact angle (surface energy) and separation efficiency results are strongly indicative of a wide range of filtration performance enhancements that can be achieved through suitably chosen surface-active modification of standard substrate materials.

## 1. Introduction

Standard filter media substrate materials exhibit filtration performance characteristics within a certain range that can visually be linked to the composition and density of the fiber matrix that results in some distribution of mean porosity within the substrate material. In turn, this distribution of porosity results in certain filtration performance characteristics usually expressed by the three main parameters of separation efficiency, differential pressure (∆P) and the dirt holding capacity (DHC) [1,2,3,4]. The usual way to further fine tune these three performance characteristics can only be accomplished by modifying the constituent fiber matrix [5,6,7,8]. Several examples of such modifications were provided in Part 1 of this paper. In contrast to such physical fiber matrix modifications, an adjustment of surface-active properties [9] to bring about an increased work of adhesion between the contaminant and the substrate has rarely been considered. We have previously carried out some background work on the topic of surface modifications in filtration applications; however, we did not directly investigate its effect on the three parameters that characterize filtration performance characteristics [10,11,12]. The novelty of this work lies in the investigation of whether it is possible to optimize filtration performance characteristics by activating the surface of the filter media with a suitably chosen surface-active agent. In addition, we wanted to investigate the effect of the oppositely treated materials, meaning how the respective hydrophilic and hydrophobic treatments would affect filtration performance.

As this second part of the overall paper will show, significant gains in filtration performance characteristics can be realized through the adjustment of material surface properties with suitably chosen surface-active agents. As the results have shown, it was possible to realize reasonably good significant gains in filtration efficiencies and pressure drops without undue penalties in dirt holding capacities. 

## 2. Materials and Methods

### 2.1. Textile Material Characterization Procedures

The filter materials used in this paper are glass fiber substrates, which consist of a nonwoven fiber mat and are used commercially for depth filtration applications. The filter media used are comparable to those used in Part 1 [13].

The prepared filter media samples were evaluated by the standard textile characterization methods that included thickness, basis weight, air permeability and the mean flow pore size (MFP), which are listed in Table 1 below [9,13,14,15,16,17,18,19].

The main method to assess the effect of the applied surface-active treatment was the contact angle measurement. The contact angle measurements were carried out on a DAS 30 instrument (Krüss GmbH, Hambur, Germany) with distilled water and diiodo methane as the polar and non-polar solvents, respectively [10].

The overall procedure for deriving the surface energy values from the contact angle measurements is illustrated in Figure 1 and expressed mathematically in Equations (1)–(3) below [10,24].
(1)$$\cos\theta =\frac{\sigma_{S}-\sigma_{LS}}{\sigma_{L}},$$
(2)$$\sigma =\sigma^{p}+\sigma^{d},$$
(3)$$\sigma_{LS}=\sigma_{S}+\sigma_{L}-2\left(\sqrt{\sigma_{S}^{p}\sigma_{L}^{p}}\sqrt{\sigma_{S}^{d}\sigma_{L}^{d}}\right),$$

In the above equations, Equation (1) is referred to as Young’s equation, Equation (2) as the Fowkes equation and Equation (3) as the Owens and Wendt equation [10,24,25,26,27,28,29,30,31,32].

The results of the contact angle measurement allow for the calculation of the total surface free energy σ of the analyzed substrate in terms of its polar and disperse components [10].

Equation (3) can also be used in reverse to calculate the component surface tensions of liquids. All that is needed is to know the surface tensions of the liquids (σ_1_ and σ_2_) and the interfacial tension between the two liquids (σ_12_). An important example would be to determine the components for the surface tension of water (liquid 1) from the measured interfacial tension between water and diiodo methane (CH_2_I_2_ = liquid 2). As diiodo methane contains no nonpolar component and is limited to only the purely dispersive, van der Waals interactions, the σ_2_^p^ = 0. Therefore, the term for the polar component drops out of Equation (3) and σ_1_^d^ can be calculated. The polar fraction for water can now be defined. The surface tension of water is due to its ability to interact through both the polar and dispersive components [26]. The properties of the liquids employed in the contact angle measurements are listed below in Table 2.

Depending on the substrate surface energy achieved through surface modification, either the sessile drop or the captive bubble method was used for the contact angle measurement [31,33]. The sessile drop measurement involved a direct liquid drop deposition on the substrate surface. This method is not suitable for strongly hydrophilic (high surface energy) substrates because the deposited water droplet sinks into the substrate nearly immediately after its deposition on the surface. The two different measurement methods are depicted diagrammatically in Figure 2 below.

The process of filtration can be thought of in two different ways. The first deals with a purely mechanical sieving action that leads to the removal of particulate matter with well-defined particle size distribution where the particles in size ranges larger than the mean flow pore size are held back because they cannot penetrate the pore. In contrast, the second mechanism is based on the attraction between the particles of certain chemical composition and the filter media substrate that results from some form of chemical attraction between them in a manner similar to a well-known principle of “like dissolves like”. The central question that arises in reference to the second mechanism therefore is to what extent the surface properties of the substrate can be tuned to increase this force of attraction between it and the particles that the constitute the suspended contaminant particles within the carrier fluid phase. In essence, the question asks if it is possible to fine tune the surface properties of the filter media substrate such that it would and/or could selectively attract and “fish out” contaminant particles of certain chemical composition more efficiently than what is dictated simply by the mean flow pore size distribution of the original material.

In order to evaluate this possibility further and to express it in more quantitative terms, a model for the interaction has to be constructed among the contaminants, the filter media and the carrier phase. The model took into account the total surface tension of each individual component, its polar and disperse components as well as their respective polarities. A typical operating environment of a multipass test stand would therefore be expressed by quantities listed below in Table 3.

SiO_2_ and Al_2_O_3_ were selected as impurities, as these are the two main components of the ISO Medium Test Dust (ISO MTD) used. SiO_2_ constitutes 68 to 76% and Al_2_O_3_ correspondingly has a percentage of 10 to 15% [35].

Material S denotes the standard glass fiber media without a surface modification. The material S-C1 denotes a hydrophilic plasma surface treatment with N_2_O and CH_4_ gas. Once the gas has been introduced into the chamber, the interior is energized to 200 W and pressurized to 50 mTorr [36]. In the liquid chemical treatments S-C2 and S-C3, the filter medium is first activated in advance using a corona pretreatment and then immersed in a liquid that surrounds the fibers. S-C2 and S-C3 denote liquid surface treatments, whereas C2 is a hydrophilic modification based on PVA/PVB and C3 is a hydrophobic modification based on fluorocarbon. After immersing the filter media in the liquids, they are then dried at 140 °C for 30 min.

The model of the work of adhesion between two materials, based on their surface properties, already lends itself well for predicting the gains in filtration performance characteristics of surface-modified filter media. For example, from the perspective of individual components expressed in a two-body framework at a certain time, it can be shown that when the work of adhesion between the filter media (F) and the contaminant (C) is larger than that between the liquid (L) and the contaminant (C), the contaminant will preferentially adhere to the filter medium: (4)$$W_{FC}>W_{LC},$$

Here, the two-body work of adhesion (W_12_ for example W_FC_) can further be broken down into its individual polar and disperse components as follows [37]:(5)$$W_{12}=2\left(\sqrt{\sigma_{1}^{p}\sigma_{2}^{p}}+\sqrt{\sigma_{1}^{d}\sigma_{2}^{d}}\right),$$

The most expedient way to validate the assumption contained in this model was to prepare several samples of filter media such that two would represent the same end of the work of the adhesion space whereas the third would lie completely opposite to it. This is best demonstrated in Figure 3 and Figure 4 below that illustrate the work of adhesion behavior for differently modified substrates and their interaction with the two contaminants, i.e., SiO_2_ and Al_2_O_3_, respectively.

In the two figures above, the white area represents the region in which the work of adhesion between each contaminant and the fluid is stronger, whereas the colored area denotes the region where the contaminants are more strongly attracted to the filter medium. Therefore, the model predicts that in the first case, the contaminants will not adhere to the filter medium and will not be effectively filtered out, whereas the opposite will be true for the second case and the contaminants will be filtered out even more efficiently than with an unmodified substrate. The diagrams depicted in Figure 3 and Figure 4 above may at first appear somewhat difficult to interpret. However, the main point is that they quite clearly illustrate the regions of interest where the interaction expressed in terms of the work of adhesion between the select contaminant and the surface-modified filter media is the greatest. This region is depicted by the green color in both figures. From the examination of both figures, it becomes clear that to remove either or both constituent contaminants of the ISO MTD test dust from the MIL-H-5606 hydraulic fluid, a filter media should be prepared such that it should exhibit a relatively high surface tension, ideally >80 mNm^—1^, and a relatively low polarity of <20%. Although this may sound like a tall order at first, upon a closer investigation of the available surface-active agents, compounds in the polyglycol class of materials, such as polyethylene glycol (PEG), would seem to fit the bill.

### 2.2. Evaluation of the Filtration Performance Characteristics

The impact of the chemical surface treatment was determined in accordance with the standard ISO Test procedures, namely, the ISO 16889:2022 [38], “Hydraulic fluid power —Filters—Multi-pass method for evaluating filtration performance of a filter element”, and the ISO 3968:2017 [39], “Hydraulic fluid power—Filters—Evaluation of differential pressure versus flow” [40]. For the multipass evaluation, a circular media sample with an effective filtration area of 176.71 cm^2^ was employed. The test was carried out with a flow rate of 7 L min^−1^ and the base upstream gravimetric (BUG) content of 8.12 mg L^−1^ of the ISO Medium Test Dust (ISO MTD). The test stand employed the standard MIL-H-5606 fluid and operated at a temperature of 40 °C at which the fluids exhibit a viscosity of 15 mm^2^ s^−1^. The test was terminated when the differential pressure across the filter sample reached 3 bar. The pressure loss characteristics of the prepared filter media samples were evaluated at viscosities of both 30 mm^2^ s^−1^ and 300 mm^2^ s^−1^ to eliminate any nonlinearity effects. The 30 mm^2^ s^−1^ evaluation was performed with Megol HLP ISO VG 32 fluid and the 300 mm^2^ s^−1^ with Megol ZB1000 fluid at temperatures of 40 °C and 57 °C, respectively. The flow rate was stepped from rest to 10 L min^−1^ in as small stepping intervals as possible (ca. 0.005 L min^−1^) or until the terminal differential pressure of 5 bar was reached.

## 3. Experimental Results

The physical textile properties of the surface-modified filter media substrate are reported in Table 4. These results confirm that the basic physical properties of the substrate have not been altered as a result of the treatment. The same can be said in regard to the mean flow pore size as illustrated in Figure 5.

Three experiments were carried out for each filter medium to confirm the repeatability of the results. The resultant mean values together with their corresponding standard deviations are shown below in Figure 6, Figure 7, Figure 8, Figure 9, Figure 10 and Figure 11. Figure 6 shows the dependence of the differential pressure on the flow rate (DPQ). Figure 7 depicts the results of the DHC obtained from the multipass test. Finally, Figure 8, Figure 9, Figure 10 and Figure 11 report the results of separation efficiencies obtained for different particle sizes for each of the surface-modified and the original unmodified substrates.

Furthermore, dirt holding capacity experiments were carried out with all four filter media using the multipass test method.

The DHC experiments also yield the separation efficiencies for particle sizes > 12 µm, >7 µm, >5 µm and >4 µm. The separation efficiency is calculated as follows:(6)$$SE=\frac{N_{b}-N_{a}}{N_{b}}\;\cdot 100,$$

N_b_ = Number of particles before the filter medium

N_a_ = Number of particles after the filter medium

In addition, the above results have been tabulated in Table 5 below. Only the mean values are reported for clarity.

## 4. Discussion

The physical textile properties reported above confirm that the substrate has remained following the surface-active modification. The gain in basis weight for the S-C2 (~+10%) and the S-C3 (~+15%) modifications is consistent with the liquid dipped method of treatment. Nevertheless, all other properties, especially the air permeability and the min, max and the mean flow pore size, remained unaltered. As these properties would be chiefly involved in governing the process of filtration, their remaining unchanged allows us to conclude that any differences in filtration performance characteristics stem from the surface modification treatment. 

The resultant effects of surface modifications of the prepared filter media substrates are summarized in Table 3 (*vide supra*). An examination of the values listed in the table allows for a direct comparison of the overlapping surface tension properties that govern the process of filtration by fine-tuning the work of adhesion. Expressing these relationships graphically as it was conducted in Figure 3 and Figure 4 provides a simple model for the determination of preferential filtration mechanisms for selected contaminants. The model can be used to explain the behavior already observed but more importantly in a predictive fashion to determine the nature of surface modifications required to fine-tune the work of adhesion towards select contaminants. 

As the diagrams in Figure 3 and Figure 4 illustrate, the optimal surface for the removal of both SiO_2_ and Al_2_O_3_ from MIL-H-5606 (a mineral petroleum oil) would need to exhibit a relatively high surface tension (>60 mN m^−1^) as well as a low polarity (<20%). Thereby, the work of adhesion would reach a relatively high value of 100 mN m^−1^ or greater and be well within the green region of the diagram.

The results of the DPQ evaluations provide clear evidence of the effect of surface modifications on filtration performance (shown Figure 6). In particular, the hydrophobic/oleophobic treatment (S-C3) exhibits a significantly higher ∆P of 0.42 bar at 10 L min^−1^ with the 30 mm^2^ s^−1^ oil compared to that of 0.26 bar for the unmodified substrate, an increase of nearly 62%. The same effect can be observed for the 300 mm^2^ s^−1^ oil where the S-C3 media sample exhibited a ∆P of 5 bar already at 5 L min^−1^ compared to just 4 bar at 10 L min^−1^ for the untreated substrate. In contrast, the two hydrophilic modifications S-C1 and S-C2 exhibit no significant differences in the ∆P values compared to the base substrate. 

In the experiments in the multipass test bench (Figure 7), all modifications achieved a lower value for dirt holding capacity compared to the base substrate S. At 3 bar, substrate S reached 2.4 g, while the two modifications S-C2 and S-C3 reached only 2.25 g, whereas the modification S-C1 only reached 2.15 g. 

As the DPQ results shown in Figure 6 illustrate, the nature of the hydrophobic/oleophobic surface of the S-C3 modified substrate presents a greater resistance to flow of the petroleum based mineral oil. As a consequence, this substrate reaches the terminal ∆P of 3 bar faster than both the unaltered material and the two hydrophilic variants, thereby resulting in a lower DHC. This is despite the fact that the S-C3 substrate exhibits significantly lower filtration efficiencies, especially for the smaller particle size ranges compared to the untreated substrate as well as the two hydrophilic variants, S-C1 and S-1C2. 

The overall filtration performance results are reported in Table 5 (vide supra) and displayed graphically in Figure 6, Figure 7, Figure 8, Figure 9, Figure 10 and Figure 11. Of particular note are the filtration efficiency results reported for the smaller particles’ size ranges displayed in Figure 9, Figure 10 and Figure 11. This is because the MFP for all filter media substrates is in the order of 12 µm and therefore the separation efficiencies for the 12 µm particles shown in Figure 8 can be thought of as being a pore size-oriented mechanical sieving action. In contrast, the results reported in Figure 9, Figure 10 and Figure 11 depict the behavior for the particles significantly smaller than the average substrate porosity. These results are especially significant in view of the fact that they also confirm the work of adhesion model developed and employed in the course of the investigation effort described in this paper. Specifically, the hydrophobic/oleophobic treated substrate exhibits the lowest separation efficiencies for particles in the <7 µm size ranges in all cases, consistent with the model presented in this paper.

The shape of the filtration efficiency curves displayed in Figure 8, Figure 9, Figure 10 and Figure 11 requires some words of explanation. Similar effects are often observed in our multipass evaluations of both the flat sheet materials as well as the assembled filter elements. Although direct observation of these phenomena is not possible, the most likely explanation is that it has to do with the displacement and/or rearrangement of the sandy contaminant cake that forms on the surface of the test element. This displacement results in a partial contaminant release from the filter and therefore manifests itself as a dip in separation efficiency. As illustrated in the figures, this effect is temporary; and after a while, the system recovers to its original equilibrium condition. We have previously observed similar effects as we described in Part 1 of this paper [13].

## 5. Conclusions

The work of adhesion filtration model developed and verified experimentally in the course of work described in this paper offers interesting possibilities for future filtration applications. The general nature of the model as well as the simplicity of its graphical representation present a convenient methodology for the selection of filtration materials with pre-defined surface energy properties. This means that once the surface energies of the corresponding materials that constitute the system are determined, the filter media substrate can be fine-tuned to enhance and optimize separation efficiencies towards the select contaminant of interest. For example, this can be oriented towards metallic or microplastic particles as well as the sandy contaminants derived from the usual environmental sources. More importantly, this approach could also be extended to include the more troublesome soft contaminants, such as the oil and/or additive degradation by-products commonly referred to as varnish. In addition, very interesting possibilities for the preparation of phase separation materials for both liquid/liquid and gas/liquid applications have now been opened [41,42,43]. The ready availability of various surface agents with highly varied surface properties in both gas and liquid forms offers a great promise for the development of radically different filtration media substrates for a great number of new applications. Although in this paper we did not explore other applications such as such as air and/or water filtration or more difficult to carry out phase separation processes, either liquid/liquid or liquid/gas, the ready availability of the rather simple to interpret work of adhesion model strongly suggests that a brand new path for investigations into the science of filtration materials has now been opened.

## Figures and Tables

**Figure 1 materials-17-02720-f001:**
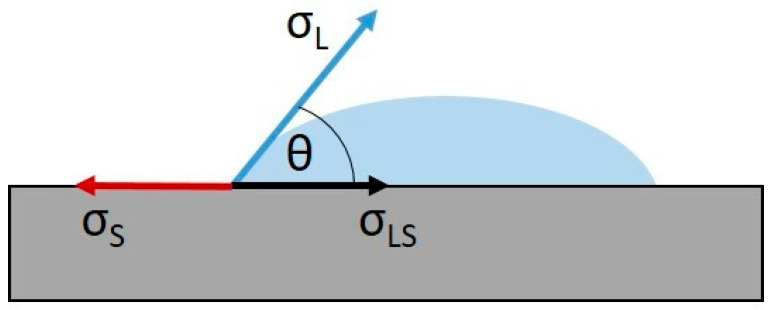
The contact angle θ results from the balance of the surface energy σ_S_, the surface tension of the liquid σ_L_ (liquid drop in blue) and the interfacial tension σ_LS_ at the three-phase boundary. (Reprinted with permission from [25]. Copyright {2024} American Chemical Society).

**Figure 2 materials-17-02720-f002:**
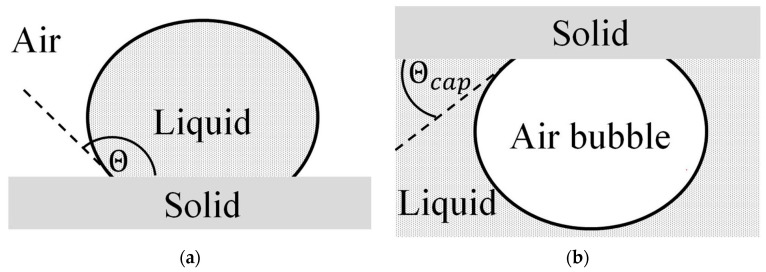
(**a**) Drop formation and contact angle θ of the “sessile drop” method and (**b**) air bubble in a liquid environment using the “captive bubble” method [11].

**Figure 3 materials-17-02720-f003:**
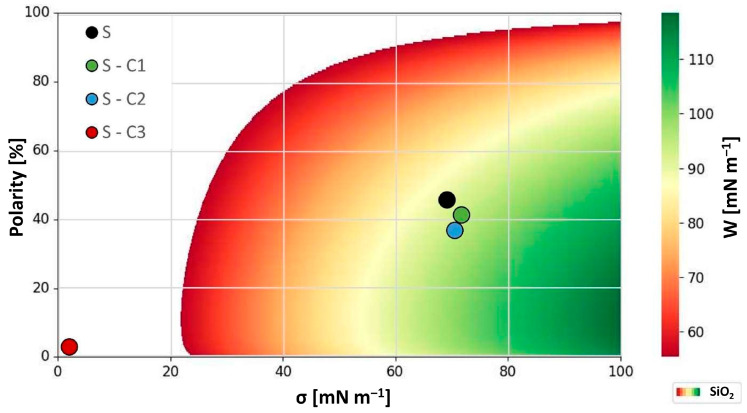
Work of adhesion diagram of MIL-H-5606 and SiO_2_ as contaminant and the filter media with corresponding surface properties.

**Figure 4 materials-17-02720-f004:**
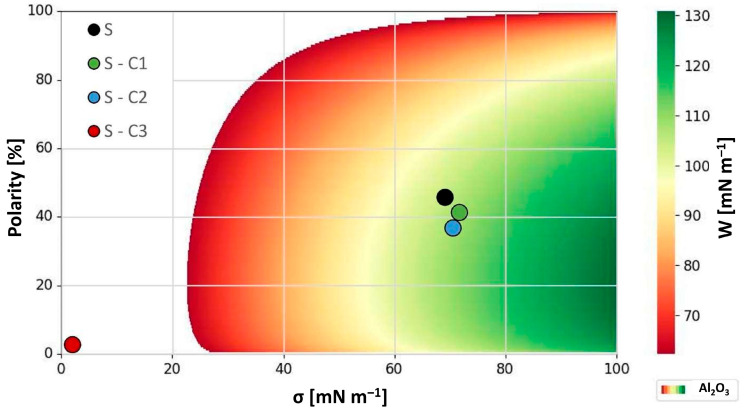
Work of adhesion diagram of MIL-H-5606 as fluid and Al_2_O_3_ as contaminant and the filter media with corresponding surface properties.

**Figure 5 materials-17-02720-f005:**
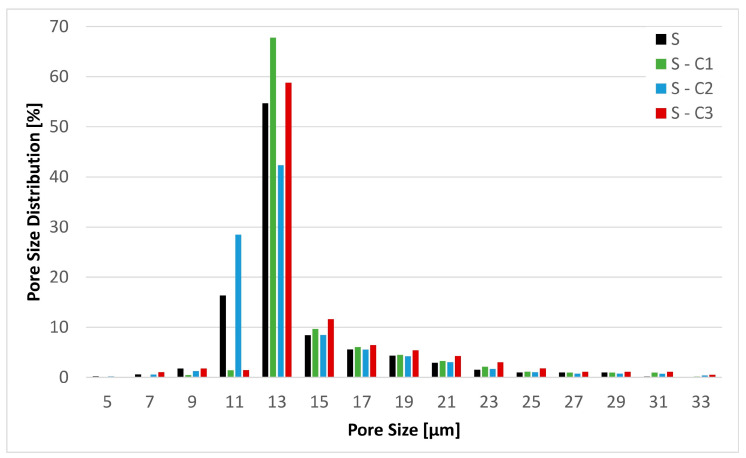
Pore size distribution of the base substrate S and the chemical modifications S-C1, S-C2 and S-C3.

**Figure 6 materials-17-02720-f006:**
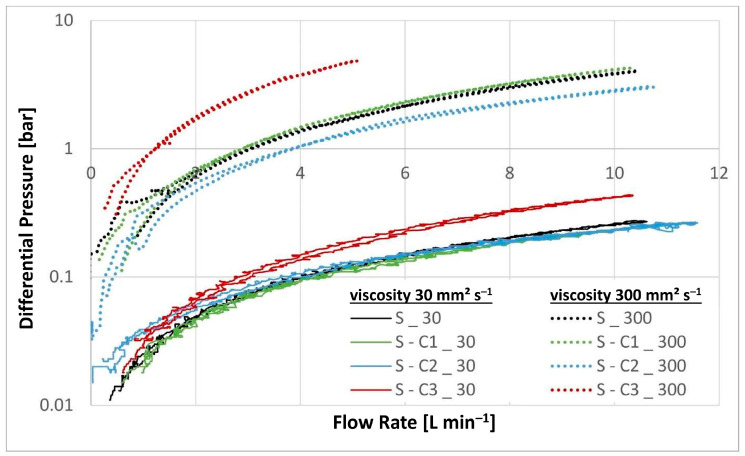
∆P results for the chemically modified nonwovens.

**Figure 7 materials-17-02720-f007:**
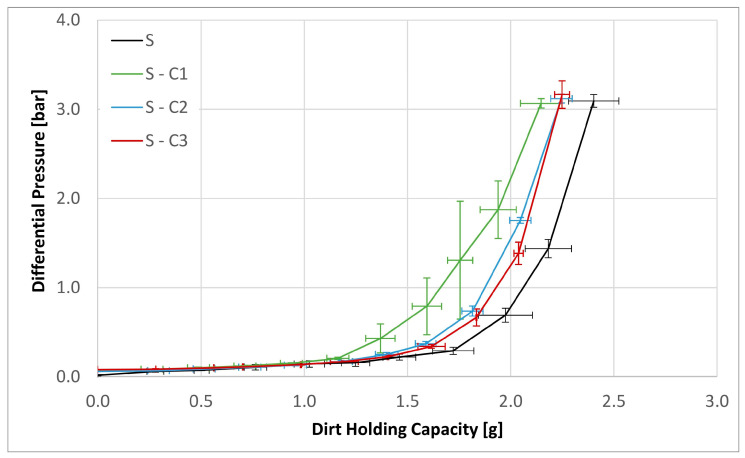
Dirt holding capacity for the base substrate S as well as the chemical modifications S-C1, S-C2 and S-C3.

**Figure 8 materials-17-02720-f008:**
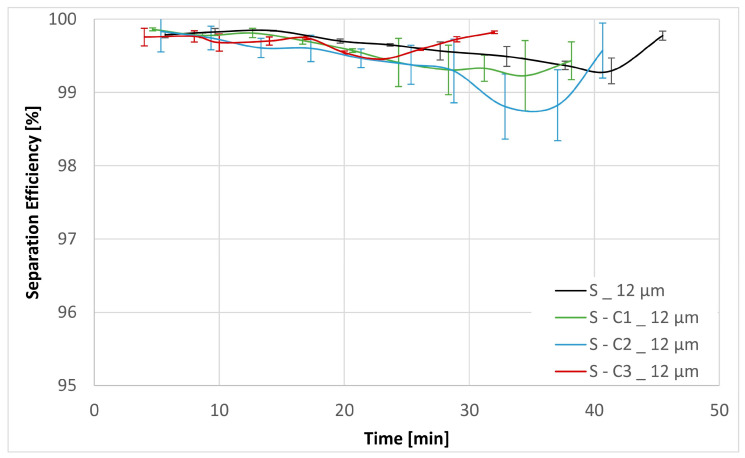
Separation efficiencies for the base substrate S as well as the chemical modifications S-C1, S-C2 and S-C3 for contamination particles with size > 12 µm.

**Figure 9 materials-17-02720-f009:**
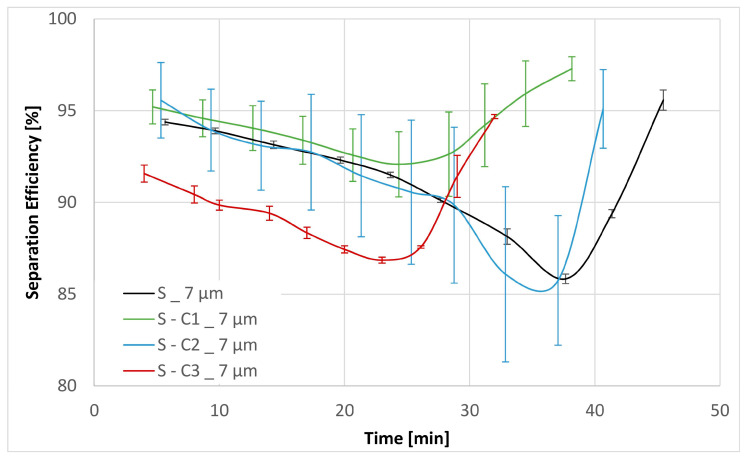
Separation efficiencies for the base substrate S as well as the chemical modifications S-C1, S-C2 and S-C3 for contamination particles with size > 7 µm.

**Figure 10 materials-17-02720-f010:**
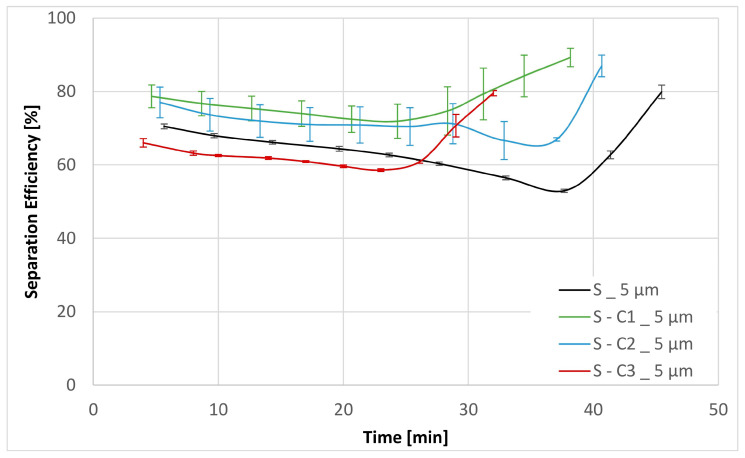
Separation efficiencies for the base substrate S as well as the chemical modifications S-C1, S-C2 and S-C3 for contamination particles with size > 5 µm.

**Figure 11 materials-17-02720-f011:**
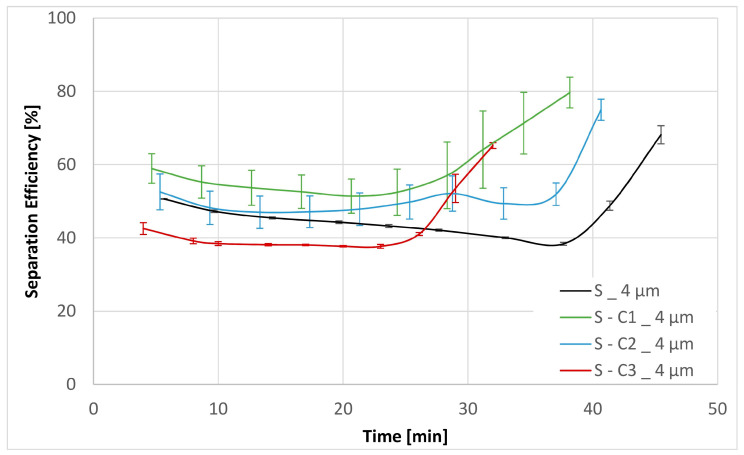
Separation efficiencies for the base substrate S as well as the chemical modifications S-C1, S-C2 and S-C3 for contamination particles with size > 4 µm.

**Table 1 materials-17-02720-t001:** Textile characterization methods and apparatus employed in the present evaluation.

Physical Properties	Method	Apparatus
Thickness	DIN EN ISO 5084:1996 [20]	Thickness gauge without pressure—Mitutoyo Deutschland GmbH, Neuss, Germany
Basis Weight	DIN EN 12127:1997 [21]	MSA225P-000-DA—Sartorius AG, Göttingen, Germany
Air Permeability	DIN EN ISO 9237:1995 [22]	FX3300—Textest AG, Schwerzenbach, Switzerland
Pore Size Distribution	ASTM F316-03:2011 [23]	Capillary Flow Porometer AX 1100—Porous Materials Inc., Ithaca, NY, USA

**Table 2 materials-17-02720-t002:** Surface energy values of fluids employed in the contact angle measurements.

Material	σ [mN m^−1^]	σ^p^ [mN m^−1^]	σ^d^ [mN m^−1^]	Polarity [%]
H_2_O	72.8	48.9	2.9	67.2
CH_2_I_2_	51.2	0.1	51.1	0.2

**Table 3 materials-17-02720-t003:** Values of the components involved in the present investigation.

Material	σ [mN m^−1^]	σ^p^ [mN m^−1^]	σ^d^ [mN m^−1^]	Polarity [%]
MIL-H-5606 [exp.]	22.7	5.7	17.0	25.2
SiO_2_ [exp.]	35.1	3.8	31.3	10.7
Al_2_O_3_ [34]	42.8	8.7	34.1	20.3
S	69.0	31.6	37.4	45.8
S-C1	71.6	29.6	42.0	41.4
S-C2	70.5	26.0	44.5	36.8
S-C3	1.9	0.1	1.8	2.7

**Table 4 materials-17-02720-t004:** Determination of physical properties of the base substrate and the chemical modifications.

Samples	Thickness [mm]	Basis Weight [g m^−2^]	Air Permeability [L m^−2^ s^−1^]	Min Pore [µm]	Max Pore [µm]	MFP [µm]
S	0.5 ± 0.0	71.7 ± 1.7	230 ± 22	5.1 ± 0.2	33.3 ± 0.1	12.8 ± 0.5
S-C1	0.5 ± 0.0	68.6 ± 0.9	237 ± 4	4.8 ± 0.1	32.3 ± 0.3	13.3 ± 0.1
S-C2	0.5 ± 0.0	78.6 ± 2.8	252 ± 5	5.2 ± 0.2	32.7 ± 0.7	12.3 ± 0.4
S-C3	0.6 ± 0.0	82.9 ± 1.7	241 ± 8	4.6 ± 0.1	33.0 ± 0.5	13.2 ± 0.3

**Table 5 materials-17-02720-t005:** Results in mean values for all experiments for the original substrate S and the chemical surface modifications S-C1, S-C2 and S-C3.

Sample	Separation Efficiency [%] for Particle Size [µm]				DHC [g]	DPQ [bar] for 30 mm^2^ s^−1^		DPQ [bar] for 300 mm^2^ s^−1^	
	>4	>5	>7	>12	3 bar	5 L min^−1^	10 L min^−1^	5 L min^−1^	10 L min^−1^
S	46.84	64.41	91.43	99.63	2.40	0.13	0.26	1.76	3.85
S-C1	59.65	77.66	94.19	99.54	2.15	0.12	0.23	1.91	4.15
S-C2	52.10	72.68	91.41	99.41	2.25	0.13	0.23	1.34	2.82
S-C3	43.17	64.39	89.76	99.68	2.25	0.19	0.42	4.84	

## Data Availability

The original contributions presented in the study are included in the article, further inquiries can be directed to the corresponding author.

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
