# Peer review of "Enhancement of Filtration Performance Characteristic of Glass Fiber-Based Filter Media, Part 2: Chemical Modification with Surface-Active Treatment"

_materials, 2024, doi:10.3390/ma17112720_

Round 1
Reviewer 1 Report
Comments and Suggestions for Authors
The manuscript presents glass fiber filter with/with coatings designs and characteristics. The tests included filtering pressure difference, separation efficiency, and filtering capacity. The results showed that, although the filtering capacity decreases and filtering pressure increases with coating, the filtering efficiency has clearly improved, especially on the small size particles. In general, the study is interesting and complete. The presentation of the manuscript needs to be significantly increased before acceptance. The detailed comments are listed below:
1. The fabrication of the coating/ surface treatment needs to be clearly stated, if possible, comparing with other surface treatment techniques on glass fiber filter as discussion.
2. The microstructure of the filter before and after surface treatments are necessary to provide.
3. Please add discussion on the uniformity of the surface treatments.
4. In section 2.2, the unit of viscosity was confusing. Please explain in detail.
5. From Fig. 3 to Fig. 11 are plotted with very low image quality. It is necessary to output high resolution clearer versions of those plots.
6. Please compare the filtering performance with the recent literature.
Author Response
Attached please find the file with our responses to your review. We would like to thank you very much for your comments and hope you will now find our paper suitable for publishing in the Materials journal. Thank you.

Reviewer 2 Report
Comments and Suggestions for Authors
Review v1
Materials
Enhancement of Filtration Performance Characteristic of Glass Fiber-Based Filter Media Part 2: Chemical Modification with Surface Active Treatment
Laura Weiter ,Stephan Leyer, John K. Duchowski
The reviewed work presents an experimental study aimed at increasing three basic properties of filter material, namely separation efficiency, differential pressure (ΔP) and dirt holding capacity (DHC) by performing modifications to selected materials. The work is within the scope of the journal Materials. The results of the research are shown in a number of figures, which allow better tracking of the results and their analysis. Interesting research and interfering results are presented, the interpretation of which should be expanded to include the interrelation of filtration efficiency and flow resistance and the DHC parameter. In the 'Conclusions' section, the authors did not provide the most important achievements from the conducted research. There is no information that responds to the goal set in the paper. The conclusions should be reformulated from scratch and supported by the results of the research. After making the indicated corrections, the work can be further processed.
Specific comments
1) In the abstract, the authors write "The aim of these modifications was to enhance the three fundamental filtration performance characteristics, namely the separation efficiency, differential pressure (ΔP) and the dirt holding capacity (DHC)." As a rule, the aim is to minimize, not increase, the flow resistance.
2) Please with the definition of the parameter "dirt holding capacity (DHC)". In Figure 7, the Authors give this parameter in the description of the horizontal axis in "g" (Dirt holding capacity [g]), which does not seem to be appropriate, because then it is not possible to compare with other studies, where there may be a different surface. At the same time, the authors refer to the paper [4], where this parameter is described Dust load [g/m2]. To confirm that this description as "Dust load" is correct I suggest the Authors use the paper: https://doi.org/10.3390/ma15207292, where the changes in filtration efficiency and flow resistance as a function of changing the "Mass loading of dust [g/m2]" are presented.
3) Under Figure 2. there is a sentence "The first deals with a purely mechanical sieving action that leads to the removal of particulate matter with well-defined particle size distribution where the particles in size ranges larger than the mean flow pore size are held back because they cannot penetrate the pore". It can be seen from the above that the filtration process in fiber beds takes part as the dominant mechanical (sieve) filtration mechanism. The basic mechanisms of filtration in the deposits are inertial, direct hooking and diffusion mechanisms.
4) Under Table 3 is the sentence, "SiO2 and Al2O3 were selected as impurities, as these are the two main components of the ISO Medium Test Dust (ISO MTD) used. SiO2 constitutes of 68 to 76% and Al2O3 correspondingly has a percentage of 10 to 15%." Indeed, these are the two main components of the test dust, but this test dust has other components. Please provide a full characterization (chemical and fractional composition) of the ISO Medium Test Dust. In the rest of the paper, the authors use SiO2 and Al2O3 impurities separately, rather than the dust as a whole. How to understand this. What about the effect on the filter medium of the other dust components.
5) The authors have carried out experimental research by giving a brief research methodology. It is advisable to supplement it with a diagram of the station and indicate its most important elements, such as dust dosing, flow resistance measurement, particle measurement.
6) Line 161-162. "…effective filtration area of 176.71 cm2 was employed." SI units should be used.
7) In formula (6), where: Nb - the number of particles upstream of the filter medium, Na - the number of particles downstream of the filter medium, are the dust particles of only SiO2 or Al2O3 taken into account or the entire spectrum of test dust.
8) The test results (filtration efficiency) shown in Figure 8-11 were performed as a function of time. How to compare these results with the flow resistance (Figure 7) performed as a function of "Dirt holding capacity". The filtration efficiency and flow resistance of the same filter material should be shown on a single graph and preferably as a function of "Dust load [g/m2]". I suggest the Authors to use the paper: https://doi.org/10.3390/ma14237166.
9) How do the Authors interpret the minimum filtration efficiency shown in the graphs of Fig. 8-11.
Author Response

(The authors gave the same response as above.)

Reviewer 3 Report
Comments and Suggestions for Authors
This manuscript presented a study about he enhancement of filtration performance characteristic of glass fiber-based filter media. The work has some potential. However, several points listed below need to be improved.
Abstract: please add more numerical results to the abstract.
Introduction: the introduction section is too short. Please better contextualize the previously works related to the main topic of this study.
Introduction: clearer the novelty of this work.
Section 2.1: I suggest at the beginning of section 2.1 better describe the textile material used in this work and the others compounds used.
Section 2.1: better describe how the surface modification (conditions, parameters,…) was done.
Section 3: I suggest better contextualize the results presentation in Section 3. Please better comment what is showed in Figure 6 to Figure 10. A brief comment about the results presented in Table 4 and Table 5 is welcome.
Section 4: I suggest add some results form the literature, and if possible, compare the results of this work with others from the literature.
Author Response

(The authors gave the same response as above.)

Round 2
Reviewer 1 Report
Comments and Suggestions for Authors
The comments were addressed.
Reviewer 3 Report
Comments and Suggestions for Authors
After corrections the manuscript reads well. I suggest publications in its current form.